# Updates and Expert Opinions on Liver Transplantation for Gastrointestinal Malignancies

**DOI:** 10.3390/medicina59071290

**Published:** 2023-07-13

**Authors:** Alexander H. Shannon, Samantha M. Ruff, Austin D. Schenk, Kenneth Washburn, Timothy M. Pawlik

**Affiliations:** Department of Surgery, Division of Surgical Oncology, The Ohio State University Wexner Medical Center, The James Comprehensive Cancer Center, Columbus, OH 43210, USA; alexander.shannon@osumc.edu (A.H.S.); samantha.ruff@osumc.edu (S.M.R.); austin.schenk@osumc.edu (A.D.S.); ken.washburn@osumc.edu (K.W.)

**Keywords:** transplant oncology, liver transplant, gastrointestinal malignancies

## Abstract

Transplant oncology is a relatively new field in which transplantation is used to treat patients who would otherwise be unresectable. New anticancer treatment paradigms using tumor and transplant immunology and cancer immunogenomics are emerging. In turn, liver transplantation (LT) has become a potential therapy for certain patients with colorectal cancer (CRC) with liver metastasis, hepatocellular (HCC), cholangiocarcinoma (CCA), and metastatic neuroendocrine tumor (NET) of the liver. Although there are established criteria for LT in HCC, evidence regarding LT as a treatment modality for certain gastrointestinal malignancies is still debated. The aim of this review is to highlight updates in the role of LT for certain malignancies, including HCC, metastatic CRC, hilar CCA, and neuroendocrine tumor (NET), as well as contextualize LT use and discuss controversies in transplant oncology.

## 1. Introduction

Surgical resection or transplantation are the only potentially curative treatment options for primary liver cancers such as hepatocellular carcinoma (HCC) and cholangiocarcinoma (CCA), as well as metastatic disease including colorectal cancer (mCRC) and gastroenteropancreatic neuroendocrine tumors of the liver. These cancers are aggressive and many patients present with advanced disease [1,2,3,4]. Consequently, a small proportion of patients may be eligible for resection at the time of diagnosis [5]. 

Liver transplantation (LT) has gained increased acceptance as a potential therapy for well-selected patients with HCC, hilar or intrahepatic CCA, and mCRC and neuroendocrine metastasis [6,7]. Transplant oncology is a relatively new field that may benefit patients with unresectable disease and allow for the study of new anticancer treatment paradigms focused on tumor and transplant immunology and cancer immunogenomics [8]. The four pillars of transplant oncology include the evolution of multidisciplinary cancer care by integrating LT, elucidating self- and non-self-recognition by linking tumor and transplant immunology, the exploration of the biomechanisms of disease via genomic studies, and extending the limits of safe surgical resection by applying transplant surgical techniques [9]. 

LT has been proposed as a mechanism to treat hepatobiliary cancers as far back as the 1970s [10,11,12,13]. Excising the entire liver en-bloc serves two purposes: it removes the primary cancer and removes the pro-carcinogenic environment from which the tumor arises [14,15]. The initial clinical studies were underwhelming and outcomes were generally poor until the Milan criteria were established for LT in HCC [16]. Since then, criteria for transplantation in the setting of malignancy have been expanded with an increasing body of evidence, suggesting efficacy in select patients, including individuals with CCA, mCRC, and neuroendocrine tumor metastases. With improved patient selection and operative techniques, as well as post-operative care, LT has become a viable curative option for many patients with hepatic and metastatic malignancies of the liver. Coupled with advances in chemotherapy and other adjuvant treatment options, the importance of LT for these disease processes has increased. We herein highlight the role of LT for HCC, hilar CCA, mCRC, and neuroendocrine tumor (NET), as well as discuss several controversies in the field of transplant oncology.

## 2. Hepatocellular Carcinoma

Hepatocellular carcinoma (HCC) is the most common primary liver cancer and a leading cause of cancer-related death worldwide [1,2,3]. Prognosis can be poor given that patients often present with advanced stage in the setting of chronic liver dysfunction or cirrhosis, precluding surgical resection [17]. Unfortunately, even among patients who undergo curative intent resection, the incidence of recurrence can be high [18]. Multidisciplinary treatment with surgical, medical, and radiation oncology is crucial. The Barcelona Clinic Liver Cancer (BCLC) guidelines provide staging criteria, prognostic information, and treatment recommendations based on tumor burden, liver function, and performance status [19]. 

### 2.1. Liver Transplant Criteria in HCC

LT offers the best chance for optimal long-term outcomes with 5-year post-LT survival of approximately 65–80% and a lifetime recurrence risk of 8–15% [20]. LT is limited by the scarcity of organs, making patient selection crucial. The Milan criteria were published in 1996 and laid the foundation for HCC LT patient selection. Patients with HCC and either a solitary lesion ≤ 5 cm or up to three lesions with each being ≤3 cm without vascular invasion or extra-hepatic involvement are candidates for LT [16]. Due to the success of LT among patients with HCC who met the Milan criteria, transplant surgeons and oncologists have worked to expand the eligibility criteria. The University of California San Francisco [UCSF] criteria recommend LT for patients with a single tumor ≤ 6.5 cm, three tumors with each being ≤4.5 cm with total tumor diameters ≤ 8 cm, or the “up to 7 criteria” where the sum of the maximum tumor diameter and number of tumors must be ≤7 cm. Similarly to the Milan criteria, the exclusion criteria for the UCSF criteria include the presence of major vascular invasion or extra-hepatic disease [21,22]. Post-transplant survival at 5 years was 80.9% and 71.2% for the UCSF and “up to 7 criteria”, respectively. 

Additional studies have continued to expand the LT criteria of eligible patients. Toso et al. applied total tumor volume (TTV) < 115 cm^3^ and alpha-fetoprotein (AFP) levels < 400 ng/mL as criteria for LT among patients with HCC. The authors demonstrated recurrence-free survival and post-transplant survival at 4 years of 68% and 74.6%, respectively [23]. The Kyoto criteria incorporated the HCC tumor marker des-γ-carboxy prothrombin (DCP) < 400 AU/mL, an abnormal variant of prothrombin, and ≤10 tumors, each <5 cm, as the criteria for LT. The authors reported 5-year overall survival (OS) of 82% and a recurrence incidence of 7% [24]. Shimamura et al. proposed the “5-5-500” criteria based on a retrospective review of 965 living donor liver transplantations [LDLTs] among patients with <5 lesions, each <5 cm, and an AFP < 500 ng/mL, of which 301 [31%] were beyond the Milan criteria; the 5-year recurrence rate was 7.3% [25]. The transplant group from Toronto proposed even broader criteria, in which patients with any number or size of tumors were potentially eligible for LT as long as there was no vascular invasion, extrahepatic disease, or poor tumor differentiation. In this study that included 210 patients who underwent transplantation for HCC from 2008 to 2012, 105 [50%] were beyond the Milan criteria. The authors reported 5-year survival comparable to patients treated within the Milan criteria (69% for beyond Milan criteria and 78% for those within Milan criteria) [26]. Table 1 summarizes the selection criteria for LT for HCC.

### 2.2. Downstaging HCC for LT

Locoregional therapies such as ablation, transarterial chemoembolization (TACE), and stereotactic body radiation therapy (SBRT) are employed to downstage patients in an attempt to make patients candidates for definitive surgical therapy (resection or transplant). In a phase IIb/III trial, 45 patients were downstaged within the Milan criteria using locoregional or systemic therapy [27]. Patients who underwent LT (*n* = 23) had 5-year OS of 77.5% versus 31.2% among patients who did not undergo LT. Locoregional therapies resulted in a complete pathologic response in the explanted liver in 23% of patients, which was associated with improved OS and disease-free survival (DFS) [28]. Conversely, patients with a poor response to locoregional therapies upon final pathology were at high risk of post-transplant recurrence [29] Response to treatment can be evaluated through imaging and tumor markers (e.g., AFP or gamma glutamyl transpeptidase) [30] Other studies have also demonstrated that a reduction in AFP to <500 ng/mL was associated with a decreased risk of HCC recurrence and improved post-transplant mortality [31].

### 2.3. Organ Availability

The finite donor pool is a rate-limiting step in transplant oncology. While the data are controversial, one way to expand the donor pool is through LDLT. The initial studies raised concerns regarding an increased risk of recurrence of HCC after LDLT; however, more recent studies have demonstrated improved 5-year OS in LDLT versus individuals who had a deceased donor [32,33,34]. The reason for this may be because patients spend less time on the wait list and are less decompensated prior to transplantation. However, LDLT should be limited to high-volume centers to minimize the risk to the donor [35]. Other options to expand the donor pool include the use of “marginal grafts” from older donors, donors after cardiac death (DCD), split livers, and hepatitis-C-infected grafts [36,37,38,39].

### 2.4. Future Directions for LT in HCC

Preventing HCC recurrence in the setting of chronic immunosuppression will be crucial to maximize the longevity of transplanted livers. Certain immunosuppression medications, such as calcineurin inhibitors, have been associated with an increased risk of HCC recurrence [40,41]. The SiLVER trial (NCT00355862) assessed the effect of sirolimus, a mammalian target of rapamycin (mTOR) inhibition, on HCC recurrence after LT. Sirolimus use for >3 months was independently associated with reduced mortality, demonstrated a benefit in OS and DFS, and decreased recurrence among patients with elevated AFP (>10 ng/mL) [42].

Equally important in the future of LT for HCC is the identification of predictors of recurrence. Established pathologic characteristics such as T-stage and histologic grade, as well as microvascular invasion and tumor markers such as AFP, have been associated with a risk of recurrence; there is also a growing body of literature regarding the use of fluorodeoxyglucose (FDG) PET/CT to predict the recurrence of HCC after LT [43,44,45,46,47,48,49]. In turn, the use of FDG PET/CT may play a role in the future for patient selection for LT. Overall, LT for HCC is a viable option with favorable outcomes for selected patients who fall within the certain criteria listed in Table 1.

## 3. Cholangiocarcinoma

Cholangiocarcinoma (CCA) is categorized into anatomical subtypes: intrahepatic and extrahepatic (hilar and distal common bile duct), each with distinct biology and behavior [50]. While surgical resection is a potential treatment option for CCA, many patients present with locally advanced CCA and are not candidates for surgery. As such, LT has been proposed as a potential curative-intent option for patients with hilar or intrahepatic CCA. Similarly to other malignancies, the initial outcomes associated with LT for CCA were underwhelming, with 5-year OS ranging from 0 to 18% [51,52]. Subsequent studies demonstrated 3- and 5-year OS of 40% and 30%, respectively, and yet the incidence of recurrence was high [53,54]. Hong et al. retrospectively compared LT versus resection among patients with locally advanced hilar or intrahepatic CCA. LT was associated with improved 5-year RFS versus resection (33% resection vs. 0% LT) [55]. These studies were limited, however, as the cohort included both hilar and intrahepatic CCA, which have distinct underlying biological natural histories. 

### 3.1. Liver Transplantation for Hilar CCA

The Mayo protocol includes the administration of neoadjuvant chemoradiation to patients with hilar CCA prior to LT. Patients with nodal disease, metastases, or tumors > 3 cm are excluded from LT consideration. Using the Mayo protocol, LT was associated with improved DFS and OS versus standard chemotherapy [56]. Gores et al. reported 5-year OS of 65–70% after neoadjuvant external beam radiation, brachytherapy, and capecitabine, followed by LT [57]. In a separate study by the European Liver and Intestine Transplant Association, 5-year OS was 59% among patients with hilar CCA who underwent LT following the Mayo protocol [57,58]. A large multicenter trial evaluating 287 patients with hilar CCA who underwent LT similarly demonstrated 5-year DFS of 65% [59]. 

Different neoadjuvant therapeutic regimens have been studied prior to LT, including gemcitabine and capecitabine with radiation [60]. There are a few retrospective analyses comparing LT to resection for hilar CCA. Ethun et al. assessed OS among patients who underwent LT versus resection for hilar CCA and noted higher 5-year OS among patients who underwent transplant (LT 64% vs. resection 18%) [61]. Hoogwater et al. reported that, while neoadjuvant chemotherapy and LT had a lower risk of tumor recurrence, there was a higher rate of post-operative vascular complications [62]. In a recent meta-analysis, neoadjuvant therapy followed by LT provided a benefit in terms of OS and tumor recurrence compared with upfront transplant [63]. Similar findings have been reported in other retrospective studies; however, there is a relative dearth of prospective studies on the topic [63,64,65,66]. There is an ongoing randomized controlled trial (TRANSPHILL, NCT02232932) comparing resection versus neoadjuvant therapy followed by LT, with results expected in the coming years.

### 3.2. Liver Transplantation for Intrahepatic CCA

Intrahepatic CCA has been associated with generally poor results following LT. The initial data were based on patients who underwent LT for a different indication and were incidentally noted regarding a small intrahepatic CCA in the explanted liver. Patients with early- or intermediate-stage intrahepatic CCA had higher median OS versus individuals with advanced disease [67]. A small cohort of 29 patients who underwent LT for early-stage intrahepatic CCA had lower rates of recurrence and improved OS at 5 years versus patients with more advanced tumors [68]. A recent study from France demonstrated that there was a survival benefit with LT versus liver resection among cirrhotic patients with early-stage intrahepatic CCA [69]. As with HCC, LT removes the underlying field defect of chronic liver disease that can contribute to the development and progression of CCA and improves overall patient health by restoring liver function. Several studies have demonstrated that neoadjuvant therapy followed by LT may confer a survival benefit compared with resection [55,70,71]. Additionally, several large retrospective analyses have confirmed the survival benefit of LT for early intrahepatic CCA [72,73]. Lee et al. noted, however, that LT for intrahepatic CCA was associated with worse OS and a higher risk of recurrence than patients undergoing LT for HCC [74]. There is a paucity of high-quality prospective data to provide evidence that LT should be adopted as a routine treatment approach for patients with intrahepatic CCA. An ongoing phase II clinical trial investigating LT in cirrhotic patients with early-stage intrahepatic CCA (NCT02878473) should help to define the role of LT. Table 2 demonstrates ongoing clinical trials related to LT for CCA. 

Future studies should focus on the selection criteria for LT for patients with CCA and identify which patients with intermediate or advanced disease are optimal candidates for LT. Genetic profiling may identify patients at higher risk of recurrence, such as individuals with mutations in *KRAS*, *BAP1*, or *CDKN2A* in intrahepatic CCA or mutations in *P53*, *BRCA1-2*, and P*IK3CA* in hilar CCA [75,76,77]. The role that these mutations play in the selection of patients for LT requires further investigation. While not as promising as HCC, LT for CCA may have a role especially combined with neoadjuvant therapy prior to transplant. 

## 4. Metastatic Colorectal Cancer of the liver 

Over the past few decades, survival among patients with CRC has improved due to improved screening modalities, aggressive surgical resection, and advancements in chemotherapy and targeted therapy [78]. However, for unclear reasons, the incidence of CRC has increased in younger populations [79,80]. Additionally, because the screening guidelines are age-dependent, CRC in the younger population is often diagnosed at a more advanced stage. Approximately 50–60% of patients with CRC will develop metastases, with the liver being the most common site [81,82].Unfortunately, 80–90% of patients will have unresectable disease, commonly due to an insufficient liver remnant [83,84]. In these patients with isolated liver metastases, LT has been proposed as a potential curative-intent option.

### 4.1. Liver Transplantation Criteria for CRC Liver Metastases

The initial data on LT for CRC liver metastases were published in the early 1990s; the long-term outcomes were poor, with low 5-year OS and a high incidence of recurrence [12,85]. Given the poor outcomes and the scarcity of grafts, LT was largely abandoned as a treatment option for metastatic CRC. Over the last decade, a Scandinavian consortium reinvigorated interest in LT for CRC liver metastases [86,87]. In a landmark study by Hagness et al., the 5-year survival following LT of 21 patients with liver-only CRC metastases who had a resected primary tumor and at least 6 weeks of chemotherapy (SECA-I study) was evaluated. The authors reported OS of 95%, 68%, and 60% at 1, 3, and 5 years, respectively. Liver tumor burden > 5.5 cm, CEA > 80 micrograms/L, and disease progression after chemotherapy were strong prognostic indicators of poor outcomes [88]. Of note, 19 of the 21 patients had tumor recurrence, but primarily as pulmonary metastases (*n* = 17) [89]. A smaller study by Toso et al. evaluated 12 patients with CRC liver metastasis who underwent LT and demonstrated DFS of 56%, 38%, and 38% at 1, 3, and 5 years, respectively [90].

Currently, LT for metastatic CRC remains somewhat controversial. Dueland et al. compared DFS, PFS, and OS among patients in the SECA-I cohort who underwent LT versus patients who received palliative chemotherapy (NORDIC VII study, *n* = 47). Although there was no difference in DFS and PFS, there was a marked difference in 5-year OS (LT cohort: 56%, versus palliative chemotherapy: 9%) [91]. It should be noted, however, that the palliative chemotherapy cohort only received first-line treatment. A follow-up study assessed OS among patients with liver-only CRC metastases who had progressed during various standard lines of chemotherapy at the time of LT. Of note, 5-year OS was 44% in this cohort, which is better than any other treatment option reported in the literature [92]. 

Using the momentum from these studies, the SECA-II study prospectively assessed patients with liver-only metastatic CRC who underwent LT. More selective criteria were employed including time from diagnosis to transplant < 1 year and at least a 10% response to chemotherapy. Using these criteria, OS was 100%, 83%, and 83% at 1, 3, and 5 years, respectively [93]. One arm of the SECA-II study examined expanded criteria for both donors and patients by including patients with resectable pulmonary metastases. Ten patients were included in the analysis with DFS and OS of 4 and 10 months, respectively [94]. Additionally, compared with patients who had HCC within the Milan criteria, individuals with low-risk CRC liver metastases (low CEA, good response to neoadjuvant therapy, low tumor burden, and short interval from primary surgery to transplant) had similar 5-year OS [95]. 

A tumor burden in the liver also demonstrated an impact on outcomes. Dueland et al. compared OS among patients who underwent LT versus individuals who had portal vein embolization (PVE) and extended liver resection for high-burden liver metastasis. Patients in the LT cohort had improved 5-year OS versus the PVE/resection cohort (45.3% vs. 12.5%, respectively) [96]. Similarly, in a separate study, LT was associated with improved survival compared with liver resection among patients with a high tumor burden [97]. Additionally, Giannis et al. reported a survival benefit of LT in a recent systematic review [98]. A recent meta-analysis from Varley et al. also concluded that there was a survival benefit for LT in the setting of non-resectable CRC metastases; however, the authors cautioned that further study was needed [99].

### 4.2. Adjuvant Chemotherapy after LT in Metastatic CRC

Adjuvant chemotherapy in the setting of tumor recurrence after LT has not been well studied. The Oslo group attempted to define the role of adjuvant therapy after LT for recurrent CRC metastases. These investigators noted that adjuvant therapy was safe and did not increase the risk of graft rejection. However, over 80% of patients reported grade 3–4 toxicity events, including pancytopenia, diarrhea, and mucositis. The authors concluded that adjuvant therapy may increase long-term survival and should be considered in the post-LT setting [100]. The role of adjuvant chemotherapy post-transplant still requires further evaluation, however. 

### 4.3. Future Directions for LT for CRC Liver Metastases

Future directions of LT for CRC metastases should investigate the staged procedure, also known as the RAPID procedure (resection and partial liver segment 2/3 transplantation with delayed total hepatectomy) [101,102]. The aim of this two-stage procedure is to perform a left lateral hepatectomy with the implantation of a left lateral segment graft. The completion of the hepatectomy is delayed, thereby allowing the growth of the graft. This approach may allow for the transplantation of smaller, partial grafts and increase the availability of organs. At this time, this technique has only been reported in case series [103]. An ongoing clinical trial is currently evaluating the two-stage approach (NCT03488953) [104]. 

The utilization of LDLT is also growing for CRC metastases. A recent study by Hernandez-Alejandro et al. demonstrated RFS and OS at 1.5 years after LDLT for CRC metastases of 62% and 100%, respectively. A low incidence of perioperative morbidity was observed for both recipients and donors [105]. A recent paper by Endo et al. noted promising results for LDLT versus deceased donor LT [106].In this study, the authors reported improved 3-year OS for LDLT versus deceased donors (66.7% vs. 45.1%, respectively), with a relatively flat hazard curve of death among patients with LDLT [106]. Jackson et al. reported a survival benefit associated with LDLT for patients with model for end-stage liver disease with sodium (MELD-Na) as low as 11, and suggested that the years of life gained were comparable or greater than with deceased donor transplantation [107]. Table 3 demonstrates currently ongoing clinical trials evaluating LT for CRC liver metastases. In conclusion, LT for mCRC shows promise and the use of living donor LT may help to expand the donor pool in this patient population. 

## 5. Neuroendocrine Tumor

LT for neuroendocrine tumor (NET) is rare and represents only 0.3% of all LT [108]. Most NET liver metastases arise from primary small bowel and pancreatic NET [109]. Given the lack of high-quality long-term data and rarity of the procedure, there is no consensus on LT eligibility criteria for this patient population [110]. Mazzaferro et al. published a set of criteria that recommended patients < 60 years old with low-grade tumors, resection of their primary tumor, metastatic disease in <50% of their liver volume, and no progression of disease after systemic therapy as the eligibility criteria for LT [111,112]. 

Although data are limited, a few studies have evaluated LT in the setting of unresectable metastatic liver NET. One study assessed 213 patients who underwent LT for metastatic NET and noted 5-year OS of 73% [113]. Pancreatic primary tumors, poor tumor differentiation, and metastatic lymph nodes were associated with poor outcomes [113]. In a separate study, Vilsteren et al. reported that primary pancreatic NET was predictive of poor outcomes after LT for metastatic NET [114]. In a separate study that compared LT versus supportive care for metastatic NET among patients who had disease over 122 months, there was an improvement in 5- and 10-year OS in the LT group (97.2% LT vs. 88.8% no transplant and 50.9% LT and 22.4% no transplant, respectively) [115]. Given that NETs are generally indolent and slow-growing, some investigators have proposed time for progression as a criterion for patient selection for transplant [116]. Despite these studies, the long-term benefits of LT for metastatic unresectable NET remain unclear, and further prospective studies are required. Overall, LT for metastatic NET is rare and further investigation to evaluate its long-term efficacy is needed. 

## 6. Expert Opinion

Transplant oncology is a quickly evolving field. While LT has been well established for HCC, controversy remains relative to the use of LT for other malignancies, and it is not currently the standard of care. Data have suggested that in appropriately selected patients, LT can prolong OS and potentially be curative. Better patient selection criteria, the identification of more accurate prognostic factors that will predict recurrence, and ongoing discernment about ethical considerations are needed. Table 4 summarizes the landmark trials involving LT for GI malignancies. 

An area of ongoing research is the investigation into prognostic factors to predict which patients may benefit and have the best outcomes from LT, especially for CRC liver metastases. The SECA-I study demonstrated that tumor diameter > 5.5 cm, CEA level > 80, time from resection of primary to LT < 2 years, and progression of systemic therapy were predictors of poor prognosis. The Fong score has been a long-used validated measure to predict the recurrence of CRC liver metastases after resection and may be useful in determining which patients have tumors that are at high risk for recurrence and therefore may not be good transplant candidates [117,118]. There are also some data that suggest that a low metabolic tumor volume (MTV < 70 cm^3^) from a FDG-PET scan is associated with better patient outcomes [119]. Other proposed prognostic factors include performance status, lymph node metastases, response to chemotherapy, and biomarkers such as ctDNA [120,121,122,123,124,125]. However, the evidence for these as true predictors of outcome is limited to retrospective studies and will need to be better elucidated with prospective data. 

Additionally, a direct comparison of liver sparing resection versus LT has not been performed. R0 resection, or microscopically margin-negative resection in which no gross or microscopic tumor is present at the resected margin, is the gold standard curative-intent treatment of patients with liver metastases or primary liver cancers [126,127]. However, among patients with CRC liver metastases with extensive disease defined as >3 metastases, 5-year OS can be <40% [126,127]. The SECA-I study demonstrated much better 5-year OS; however, these patients were purposefully selected, and so direct comparisons are not feasible [128]. There is also some debate about the role of LT in patients with borderline resectable disease, as this would increase the number of patients on the wait list for grafts [129]. Additionally, there are no high-quality data directly comparing locoregional therapies such as hepatic artery infusion pumps, Y90, and transarterial chemoembolization (among others) to LT, relative to oncologic outcomes.

The role of immunosuppression in post-transplant is also another understudied area that requires further study as LT becomes increasingly adopted to treat malignancies. Many of the existing data surround the use of mTOR inhibitors, but the role of other more common immunosuppressive regimens such as calcineurin inhibitors has not been extensively studied in terms of graft survival and long-term outcomes. Future studies will need to focus on how immunosuppression impacts tumor recurrence, given the role that the immune microenvironment plays in tumor development and progression in the liver [15].

In addition, there are ethical considerations related to transplanting organs for malignancy. The wait list for organs is already long and adding cancer patients as candidates will only increase the wait time for patients with non-oncologic indications. It is imperative that outcomes for transplant oncology recipients be comparable to those of non-oncologic patients. If oncology patients are added to the wait list, how will they be prioritized on the wait list? HCC and hilar CCA patients receive MELD exception points to increase their priority on wait lists [130,131,132,133]. However, no such exception points are currently made available for other types of malignancies. As the data continue to grow in support of LT for oncologic indications, the need to prioritize these patients versus other patients on the list will need to be discussed to prevent increased mortality on the wait list. Methods to increase the donor pool will become crucial, such as LDLT, the RAPID procedure, and the use of marginal deceased donor grafts. The SOULMATE trial (NCT04161092) is a randomized study assessing LT with higher-risk allografts in non-resectable CRC liver metastases that aims to decrease the risk of long wait times for patients on the wait list [134]. 

## 7. Conclusions

The field of transplant oncology has increased rapidly over the last few decades in terms of LT for primary and secondary liver cancer. LT for oncologic indications has quickly become an increasingly viable option to treat patients and has been associated with favorable long-term outcomes. LT is important to consider as it provides a possible treatment to patients who may not have other viable options. However, expanding LT to patients with malignant indications may strain organ allocation, and the donor pool will need to be increased through the RAPID procedure and LDLT. Depending on the results of upcoming clinical trials, LT for oncologic indications will likely become more common in the future. 

## Figures and Tables

**Table 1 medicina-59-01290-t001:** Criteria for liver transplant in patients with hepatocellular carcinoma.

Criteria	Year Published	Definition
Milan Criteria	1996	Single tumor < 5 or 3 tumors < 3 cm.
University of California, San Francisco, Criteria	2001	Single tumor < 6.5 cm or 3 tumors < 4.5 cm, with total diameter of tumors < 8 cm.
Total Tumor Volume Criteria	2008	Total tumor volume < 115 cm^3^ and serum AFP < 400 ng/mL.
Up to 7 Criteria	2009	Total diameter of all tumors < 7 cm and number of tumors < 7.
Kyoto Criteria	2013	Number of tumors < 10, largest tumor < 5 cm, and DCP < 400 mAU/mL.
Toronto Criteria	2016	No size or number of tumor cut off. Must not have vascular invasion, extrahepatic disease, or poor differentiation.
5-5-500 Rule	2019	Size of tumor < 5 cm, number of tumors < 5, and AFP < 500 ng/mL.

Abbreviations: AFP: alpha-fetoprotein; DCP: des-gamma-carboxyprothrombin.

**Table 2 medicina-59-01290-t002:** Ongoing clinical trials evaluating liver transplant in the treatment of cholangiocarcinoma.

Trial Name	Start Date	End Date	Enrollment	Treatment	Patient Population	Primary End Point
NCT04378023	2020	2025	34	Neoadjuvant chemo-radiation + LT	Unresectable Hilar CCA	1-, 3-, 5-year OS
NCT02878473	2018	2029	30	LT	Early Intrahepatic CCA	5-year patient survival
NCT04556214(TESLA)	2020	2035	15	LT	Unresectable Intrahepatic CCA	3-year OS
NCT04993131(TESLA II)	2021	2035	15	LT	Unresectable Perihilar CCA	3-year OS

Abbreviations: LT: liver transplant; CCA: Cholangiocarcinoma; OS: overall survival.

**Table 3 medicina-59-01290-t003:** Ongoing clinical trials evaluating liver transplant in the treatment of colorectal cancer liver metastases.

Trial Name	Start Date	End Date	Enrollment	Treatment	Patient Population	Primary End Point
NCT04161092 (SOULMATE)	2020	2029	45	LT vs. best alternative care	Non-resectable CRC Liver metastases	5-year OS
NCT02597348 (TRASMET)	2016	2026	94	LT + chemotherapy vs. chemotherapy alone	Non-resectable CRC Liver metastases	5-year OS
NCT02864485	2016	2025	20	LDLT	Non-resectable CRC Liver metastases	5-year patient survival and DFS
NCT05248581	2019	2027	25	LT or LDLT	Non-resectable CRC Liver metastases	5-year RFS and OS
NCT05175092	2023	2030	50	NT + LDLT	Non-resectable CRC Liver metastases	5-year OS
NCT038003436	2019	2024	22	LT vs. chemotherapy	CRC liver metastases	5-year OS
NCT03494946 (SECAIII)	2016	2027	30	LT vs. chemotherapy, TACE, or SIRT	Non-resectable CRC Liver metastases	2-year OS
NCT04874259	2022	2026	20	LDLT	CRC liver metastases without prior treatment	1-year patient survival
NCT03488953 (Liver Two Heal)	2018	2023	40	LDLT with two-staged hepatectomy	Non-resectable CRC Liver metastases, stable disease, or regression after NT	3-year OS
NCT05186116 (LIVERMORE)	2022	2027	25	LDLT	Non-resectable CRC Liver metastases	5-year OS and DFS
NCT02215889	2014	2028	20	Partial liver segment 2/3 transplantation	Non-resectable CRC Liver metastases	% of patients receiving second-stage hepatectomy

Abbreviations: LT: liver transplant; CRC: colorectal cancer; OS: overall survival; LDLT: living donor liver transplant; DFS: disease-free survival; RFS: recurrence-free survival; NT: neoadjuvant therapy; TACE: transarterial chemoembolization; SIRT: selective internal radiation therapy.

**Table 4 medicina-59-01290-t004:** Summary of landmark studies involving LT for GI malignancies.

Malignancy	Landmark Studies	Findings
HCC	Mazzaferro et al. [16]	Indications for LT in patients with HCC. Single tumor < 5 or 3 tumors < 3 cm.
CCA	De Vreede et al. [56]	Improved DFS and OS for LT + NT vs. standard chemotherapy.
	Gore et al. [57]	NT radiation, brachytherapy, and capecitabine + LT, 5-year OS: 65–70%.
CRC	Hagness et al. [88]	LT + NT, 1-year OS: 95%, 5-year OS: 60%.
	Dueland et al. [91]	LT vs. palliative chemotherapy, 5-year OS: 56%. LT vs. 9% chemotherapy.
	Dueland et al. [92]	More selective criteria for LT, time to transplant < 1 year, 10% response to chemotherapy, 1-year OS: 100%, 5-year OS: 83%.
NET	Mazzaferro et al. [111]	Selection criteria for LT, <60 years old, low-grade tumors, resection of primary tumor.
	Le Treut et al. [113]	LT in unresectable metastatic NET, 5-year OS: 73%.

Abbreviations: LT: liver transplant; GI: gastrointestinal; HCC: hepatocellular carcinoma; CCA: cholangiocarcinoma; NT: neoadjuvant therapy; DFS: disease-free survival; OS: overall survival; CRC: colorectal cancer; NET: neuroendocrine tumor.

## Data Availability

Data sharing not applicable.

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
