# Peer review of "Updates and Expert Opinions on Liver Transplantation for Gastrointestinal Malignancies"

_medicina, 2023, doi:10.3390/medicina59071290_

Round 1

Reviewer 1 Report

The review manuscript entitled “Updates and Expert Opinion in Liver Transplantation for Gastrointestinal Malignancies”, discusses very hot topic of the last decade namely transplant oncology. It elaborates about the role of the liver transplantation in treatment of patients with hepatocellular (HCC), cholangiocarcinoma (CCA), colorectal cancer (CRC) with liver metastasis and metastatic neuroendocrine tumor (NET) to the liver.

           Congrats. No comments.

Author Response

Response to Reviewer 1

The review manuscript entitled “Updates and Expert Opinion in Liver Transplantation for Gastrointestinal Malignancies”, discusses very hot topic of the last decade namely transplant oncology. It elaborates about the role of the liver transplantation in treatment of patients with hepatocellular (HCC), cholangiocarcinoma (CCA), colorectal cancer (CRC) with liver metastasis and metastatic neuroendocrine tumor (NET) to the liver.

           Congrats. No comments.

Response to Reviewer 1: Thank you for your comments and your favorable review.

Reviewer 2 Report

The authors share updates on the utilization of Liver transplantation (LT) for various oncology malignancies and indicate the barriers and success rate of liver transplantation along with resection strategies.

The review structure is well organized. The introduction and the conclusion are reasonable, given the article's premise.

However, there are following suggestions for the authors to address

  1. Introduction: The closing paragraph needs additional text to highlight LT's importance better. 
  2. Results/Subheaders for LT and different tumors: The authors should add a conclusion line (main takeaway)for each section should be added to each subsection.
  3. Conclusion: The authors need to reword/add line/s in the conclusion for the importance and future of LT in various malignancies.  
  4. Since this review has a lot of information, authors should consider a graphical representation highlighting the summary of the review.
  5. Line 339: Expand on what RO resection means. 
  6. There are several incidences where wrong/ grammatically wrong words have been used in the review, including those indicated below. The authors should correct the words and highly recommend reviewing the document in detail for grammatical/wrong use words.
    1. Line 29: replace or with "for" in the line options "or" primary liver
    2. Line 47: replace "series" with "studies" in the line Initial clinical "series" were
    3. Line 375: replace "stain" with "strain" in the line may "stain" organ allocation

The English writing needs a grammatical review. Several words are not correct and have been included in the author's comments.

Author Response

Reviewer 2

The authors share updates on the utilization of Liver transplantation (LT) for various oncology malignancies and indicate the barriers and success rate of liver transplantation along with resection strategies.

The review structure is well organized. The introduction and the conclusion are reasonable, given the article's premise.

However, there are following suggestions for the authors to address

  1. Introduction: The closing paragraph needs additional text to highlight LT's importance better. 
  2. Results/Subheaders for LT and different tumors: The authors should add a conclusion line (main takeaway)for each section should be added to each subsection.
  3. Conclusion: The authors need to reword/add line/s in the conclusion for the importance and future of LT in various malignancies.  
  4. Since this review has a lot of information, authors should consider a graphical representation highlighting the summary of the review.
  5. Line 339: Expand on what RO resection means. 
  6. There are several incidences where wrong/ grammatically wrong words have been used in the review, including those indicated below. The authors should correct the words and highly recommend reviewing the document in detail for grammatical/wrong use words.

Line 29: replace or with "for" in the line options "or" primary liver

Line 47: replace "series" with "studies" in the line Initial clinical "series" were

Line 375: replace "stain" with "strain" in the line may "stain" organ allocation

Response to Reviewer 2: Thank you for your comments. Your comments have been addressed below and will enhance the paper.

  1. Additional text has been added to the closing paragraph of the introduction (lines 50-55).
  2. We have added concluding statements for each subsection (line 142-143, lines 207-209, lines 299-300, and lines 327-328).
  3. We have added a line in the conclusion regarding the importance of LT in treatment of malignancies (line 394-395) and address the future of LT in lines 397-398.
  4. We have added Table 4 which summarizes the landmark trials discussed in the text (line 332)
  5. The definition of R0 resection has been added (lines 358-359).
  6. These errors have been fixed.

Reviewer 3 Report

The manuscript is well written and about an evolving specialty. I found it interesting and understandable. The use of the clinical trial data was clear and compelling.

Author Response

Response to Reviewer 3

The manuscript is well written and about an evolving specialty. I found it interesting and understandable. The use of the clinical trial data was clear and compelling.

Response to Reviewer 3: Thank you for your comments and favorable review.